# Impact of Pomegranate Juice on the Pharmacokinetics of CYP3A4- and CYP2C9-Mediated Drugs Metabolism: A Preclinical and Clinical Review

**DOI:** 10.3390/molecules28052117

**Published:** 2023-02-24

**Authors:** Kenza Mansoor, Razan Bardees, Bayan Alkhawaja, Eyad Mallah, Luay AbuQatouseh, Mathias Schmidt, Khalid Matalka

**Affiliations:** 1Department of Pharmaceutical Medicinal Chemistry and Pharmacognosy, Faculty of Pharmacy and Medical Sciences, University of Petra, Amman 11196, Jordan; 2Herbresearch Germany, 86874 Mattsies, Germany; 3Matalka’s Scientific Writing, Burlington, MA 01803, USA

**Keywords:** cytochrome P450 (CYP450, CYP3A and CYP2C9), pharmacokinetics, pharmacodynamics, pomegranate juice

## Abstract

The *Punica granatum* L. (pomegranate) fruit juice contains large amounts of polyphenols, mainly tannins such as ellagitannin, punicalagin, and punicalin, and flavonoids such as anthocyanins, flavan-3-ols, and flavonols. These constituents have high antioxidant, anti-inflammatory, anti-diabetic, anti-obesity, and anticancer activities. Because of these activities, many patients may consume pomegranate juice (PJ) with or without their doctor’s knowledge. This may raise any significant medication errors or benefits because of food-drug interactions that modulate the drug’s pharmacokinetics or pharmacodynamics. It has been shown that some drugs exhibited no interaction with pomegranate, such as theophylline. On the other hand, observational studies reported that PJ prolonged the pharmacodynamics of warfarin and sildenafil. Furthermore, since it has been shown that pomegranate constituents inhibit cytochrome P450 (CYP450) activities such as CYP3A4 and CYP2C9, PJ may affect intestinal and liver metabolism of CYP3A4 and CYP2C9-mediated drugs. This review summarizes the preclinical and clinical studies that investigated the impact of oral PJ administration on the pharmacokinetics of drugs that are metabolized by CYP3A4 and CYP2C9. Thus, it will serve as a future road map for researchers and policymakers in the fields of drug-herb, drug-food and drug-beverage interactions. Preclinical studies revealed that prolonged administration of PJ increased the absorption, and therefore the bioavailability, of buspirone, nitrendipine, metronidazole, saquinavir, and sildenafil via reducing the intestinal CYP3A4 and CYP2C9. On the other hand, clinical studies are limited to a single dose of PJ administration that needs to be protocoled with prolonged administration to observe a significant interaction.

## 1. Introduction

In the last two decades, green medicine, such as herbs, fruits, and dietary supplements for disease management, has increased globally due to their safety, lower price, and natural components [1,2]. However, many drugs exhibited significant changes in their pharmacokinetic or pharmacodynamics properties when co-administrated with food, herbs, dietary supplements, drugs, beverages, or others [3,4,5]. Hence, drug-food interactions became the focus of healthcare professionals [6].

With co-administration of the drugs with herbal or dietary supplements, there is a potential risk of causing undesired adverse drug effects or even toxicity or reduction of the desired effect of the prescribed drug, which could ultimately lead to unsuccessful treatment. Drug interactions could affect the pharmacokinetic parameters, including absorption, distribution, metabolism, and elimination, and/or the pharmacodynamics properties, including changes in the drug effect or toxicity. Moreover, food-drug interactions might have caused significant medication errors, such as the well-known grapefruit interactions. Thus, the growing wave of green medicine and replacement therapy, local beverages, and unaddressed natural substances have imposed health concerns and should be spotlighted.

The *Punica granatum* L. (pomegranate) fruit juice contains large amounts of polyphenols, mainly tannins and flavonoids. Of these tannins is an ellagitannin, which is hydrolyzed to ellagic acid, punicalagin and punicalin. As for flavonoids, pomegranate juice (PJ) contains anthocyanins, flavan-3-ols, and flavonols. All the latter constituents have high antioxidant activity and anti-inflammatory functions [7]. For instance, polyphenols have high antioxidative potential and were found to present anti-atherogenic, anti-hypertensive, and anti-inflammatory effects. Furthermore, pomegranate has anti-bacterial, anti-parasitic, and anti-fungal activities [8,9,10].

It has been shown that pomegranate fruit constituents also have anti-cancer activities, such as inhibiting cancer cell growth and inducing apoptosis in human prostate, breast, lung, and skin cancer cells. In addition, fruit extracts have been shown to inhibit several signaling pathways, such as mitogen-activated protein kinases (MAPK) PI3K/Akt and NFκB. Based on in-vitro and preclinical studies, clinical studies were implemented using PJ in cancer patients such as prostate cancer [11].

Although PJ has been suggested to have strong anti-inflammatory, antioxidant, anti-obesity, anti-diabetic, and anti-tumoral properties, many patients may be consuming PJ with or without their doctor’s knowledge; the European Food Safety Authority’s (EFSA) position has been made clear. Many health claims have been rejected due to the fact that no sufficient cause-and-effect relationship between the consumption of pomegranate-derived products and health beyond basic nutrition is well-established [12]. Thus, the question would be whether there are risks from consuming PJ with specific types of drugs.

It has been shown that pomegranate constituents inhibit cytochrome P450 (CYP450) activities such as CYP3A [13,14] and CYP2C9 [15,16]. Therefore, pomegranate may affect the intestinal and liver metabolism of CYP3A and CYP2C9-mediated drugs. This review summarizes the preclinical and clinical studies that investigated the impact of PJ on the pharmacokinetics (PK) of drugs that are metabolized by CYP3A and CYP2C9 to recognize the risk of interactions between pomegranate intake and such drugs. Therefore, it will serve as a future road map for researchers and policymakers in the fields of drug-herb, drug-food and drug-beverage interactions.

## 2. Methodology

Our focus was on the preclinical and clinical studies that investigated the impact of PJ on the PK of drugs that are metabolized by CYP3A and CYP2C9. Therefore, our literature research in electronic databases (PubMed, Scopus, Google, Science Direct) included the following search phrases: “pomegranate and drugs”, “pomegranate and drugs pharmacokinetics”, “pomegranate and CYP3A”, and “pomegranate and CYP2C9”. In addition, any study that was not on PK or PK-pharmacodynamics (PD) related was excluded.

## 3. Preclinical Studies: PJ Impact on the Pharmacokinetics of CYP3A4- and CYP2C9- Mediated Drugs Metabolism

The preclinical studies showing the impact of PJ on the pharmacokinetics of drugs metabolized by CYP3A4 and CYP2C9 are summarized in Table 1.

### 3.1. Pomegranate and Carbamazepine

Carbamazepine is an anticonvulsant medication for treating seizures, nerve pain and bipolar disorders. Carbamazepine metabolism is mediated by CYP3A [24]. In rats, a single dose of PJ increased the area under the curve (AUC) and the highest concentration (C_max_) of carbamazepine and its metabolite, carbamazepine-10,11-epoxide, compared to water-fed rats [13]. The change in PK parameters was similar to grapefruit juice-administered rats [13]. In-vitro, increasing doses of PJ inhibited human CYP3A by determining the activity of carbamazepine-10,11-epoxide, and also increasing the preincubation time resulted in inhibiting carbamazepine-10,11-epoxide activity [13]. These investigators also observed that the time of CYP3A recovery from a single dose of PJ would be approximately 3 days. These data suggest that a single dose of PJ reduces the intestinal but not hepatic CYP3A metabolism of carbamazepine in rats.

### 3.2. Pomegranate and Tolbutamide

Tolbutamide is a sulfonylurea that is used in type-2 diabetes to stimulate insulin secretion. Tolbutamide is a substrate for CYP2C9. In rats, a single dose of PJ increased the tolbutamide AUC 1.2-fold but did not modulate the elimination half-life [15]. These data suggest that a single dose of PJ inhibits the intestinal but not the hepatic CYP2C9 metabolism of tolbutamide in rats.

### 3.3. Pomegranate and Buspirone

Buspirone is an anxiolytic drug, and it is metabolized by CYP3A. In rabbits following repeated dosing of PJ for 7 days, the buspirone AUC and C_max_ increased five-fold, and the elimination half-life increased 1.3-fold compared to buspirone alone [17]. These data suggest that repeated administration of PJ affects intestinal and liver CYP3A metabolism of buspirone in rats.

### 3.4. Pomegranate and Nitrendipine

Nitrendipine is a calcium channel blocker and is used primarily to treat hypertension. The effect of PJ on the PK of nitrendipine was studied in a single-pass intestinal perfusion model and preclinically in rodents. Rabbits pretreated with PJ for 7 days exhibited higher nitrendipine AUC and C_max_ values but did not show an increased elimination half-life compared to control-treated rabbits [18]. However, a single co-administered dose of PJ did not alter the PK parameters of nitrendipine. In rats, PJ administration as a single co-administered dose or 7-day administration increased the AUC and C_max_ of nitrendipine, but not the elimination half-life, by around two- and five-fold for the AUC and 1.4- and four-fold for the C_max_, respectively [19]. Since nitrendipine is metabolized in the liver by CYP3A4 [25], the above results suggest that PJ reduces intestinal, but not hepatic, metabolism of nitrendipine in rats. Furthermore, using a single-pass intestinal perfusion model, PJ increased permeability, absorption rate constant and nitrendipine absorption. The latter results suggest that PJ enhances the absorption of drugs mediated by P-glycoprotein, such as nitrendipine [26].

### 3.5. Pomegranate and Metronidazole

Metronidazole is an anti-protozoal and antibiotic drug currently used in intestinal, gynecologic and dental infections and others [27]. Multiple dosing, but not a single dose, of PJ, increased C_max_ and AUC of metronidazole by 1.4 and 2.3 folds, respectively, in rats [20]. Metronidazole gets metabolized primarily by CYP2A6, but CYP3A4 plays some role in 2-hydroxymetronidazole formation [28]. Thus, it would be expected that a single PJ may have no effect on metronidazole metabolism, but multiple doses might increase the bioavailability of metronidazole.

### 3.6. Pomegranate and Sildenafil

Sildenafil is a drug prescribed for the treatment of erectile dysfunction and pulmonary hypertension. In three clinical cases, Senthikumaran et al. (2012) reported that taking PJ with sildenafil prolonged erection episodes beyond orgasm [29]. Furthermore, Mallah *et al.* showed that PJ increased sildenafil bioavailability in rats in a dose-dependent fashion. The area under the curve (AUC) was increased when a higher amount of PJ was given to rats. In addition, T_max_ was delayed, and the elimination rate of sildenafil was decreased when combined with pomegranate [21].

Sildenafil is metabolized mainly by CYP3A4 (79%) and, to a lesser extent, by CYP2C9 (19%) [30]. Since pomegranate constituents impair CYP2C9, this may be the main reason for increasing sildenafil bioavailability and therefore prolonging the erection period. In addition, the delay in absorption may be related to pomegranate interaction with CYP3A4.

### 3.7. Pomegranate and Saquinavir

Saquinavir is a retroviral protease inhibitor used for human immunodeficiency virus infection treatment. Saquinavir is metabolized by CYP3A4 in the gastrointestinal tract and liver, and also its absorption in the intestinal mucosa is mediated by the efflux transporter P-glycoprotein [31,32].

In a single-dose treatment of pomegranate and saquinavir, PJ increased saquinavir’s AUC and C_max_ at the three doses of pomegranate. Also, the elimination half-life was increased. In an ex vivo model, PJ also increased saquinavir transport to the mucosal compartment in a time-dependent manner similar to P-glycoprotein known inhibitors [22]. However, following the 15-day administration of saquinavir and PJ, the AUC and C_max_ parameters were reduced in pomegranate-treated rats in addition to delaying in T_max_ and reduction in elimination half-life elimination [22]. In the latter experimental, however, both pomegranate and saquinavir were administered for 15 days, i.e., two variables were introduced. Therefore, it would be difficult to interpret the PJ effects following prolonged administration since multiple dosing of saquinavir also increased the bioavailability of midazolam, a substrate for CYP3A4 in healthy volunteers [33]. In the latter clinical study, 2 weeks of saquinavir increased midazolam’s AUC, C_max_, and elimination half-life several-fold and reduced its circulatory metabolite, 1’-hydroxymidazolam [33]. The latter study demonstrated that when both CYP3A4-mediated drugs are administered, their bioavailability will be modulated.

### 3.8. Pomegranate and Warfarin

Warfarin is widely used as an anticoagulant. As a narrow therapeutic index drug, interactions with juices should be addressed. A case study reported that PJ was the reason for having a stable international normalized ratio (INR) in a patient who took warfarin for several months [34]. When the patient stopped taking PJ, her INR became subtherapeutic. In rats, when pomegranate peel extract with 40% ellagic acid was combined with warfarin, INR was increased, but without affecting warfarin pharmacokinetics [23]. On the other hand, ellagic acid administration increased the C_max_ of warfarin [23]. In another animal study on PJ interaction with warfarin PK, pomegranate intake increased prothrombin time and INR [35]. Warfarin is present in different enantiomers (R and S), and each was found to be metabolized by different sets of enzymes. For instance, R-warfarin is metabolized primarily by CYP1A2, whereas S-warfarin is metabolized primarily by CYP2C9 [36]. Since pomegranate was found to inhibit CYP2C9, then it would be expected that pomegranate increases the INR in warfarin-treated patients.

## 4. Preclinical Studies: PJ Impact on Drugs Not Metabolized by CYP3A4 and CYP2C9

In this section, the preclinical studies that investigated the impact of PJ on the pharmacokinetics of drugs that are not metabolized by CYP3A4 and CYP2C9 are presented in Table 2.

### 4.1. Pomegranate and Metformin

Metformin is mainly used in diabetes mellitus to reduce blood glucose levels. The interaction between pomegranate and metformin was studied by Awad et al. This study demonstrated that pre-administration of PJ decreased metformin C_max_ but not AUC [37]. It has been demonstrated that metformin inhibits glycoprotein expression by inhibiting transcriptional factor, nuclear factor-kappa B, which suggests that the known pomegranate interference with glycoprotein function is not the reason for pomegranate reducing metformin C_max_ [40]. Furthermore, since metformin is metabolized by CYP2C11, 2D1, and 3A1 in rats [41], then the study above indicates that PJ may not interact with the latter enzymes.

A clinical study investigating the effect of PJ intake for 8 weeks on type-2 diabetic patients on metformin found that drinking PJ for 8 weeks significantly lowered fasting blood sugar, insulin and insulin resistance compared to patients who did not drink PJ [42]. The latter clinical study and other studies (reviewed by [43], showed the importance of PJ for diabetic patients and suggested that PJ has a significant additive effect with medications to control diabetes.

### 4.2. Pomegranate and Piracetam

Piracetam is a nootropic drug that enhances mental performance by improving cognitive functions and memory [44]. Eighty to 100% of piracetam is excreted unchanged in the urine, and therefore you would not expect to observe an effect of PJ administration on piracetam PK parameters [38].

### 4.3. Pomegranate and Theophylline

Theophylline, a well-known narrow therapeutic index drug, is mainly used for treating asthma and chronic obstructive pulmonary disease (COPD). PJ interaction with theophylline was studied in rats where the juice was added to drinking water 12 h before the experiment, and a dose was given 30 min before theophylline administration. No significant effect was observed on PK parameters such as AUC, C_max_, and T_max_ [39]. Theophylline is metabolized by CYP45 microsomal enzymes, mainly by CYP1A2 and, to a lesser extent, by CYP2E1 [45]; thus, PJ may not interfere with those enzymes.

## 5. Clinical Studies: PJ Impact on CYP3A4- and CYP2C9-Mediated Drugs Metabolism

The clinical studies investigating the impact of PJ on the pharmacokinetics of drugs metabolized by CYP3A4 and CYP2C9 are presented in Table 3.

Midazolam is a benzodiazepine medication used for sedation and anxiety. It is metabolized by CYP450 enzymes and glucuronide conjugation. Although both PJ and grapefruit juice inhibited CYP3A in vitro, Farkas et al. (2007) could not observe an effect of PJ on the clearance of oral or intravenous midazolam [14]. The main in vitro difference in the inhibition of CYP3A between PJ and grapefruit juice was that preincubation with grapefruit decreased the IC_50_, whereas the IC_50_ increased in the case of PJ, suggesting that PJ does not inhibit CYP3A (triazolam hydroxylation) in mechanism-based action [14]. On the other hand, as mentioned earlier, PJ inhibited human CYP3A4 by determining the activity of carbamazepine-10,11-epoxide formation and also increasing the preincubation time, which resulted in inhibiting carbamazepine-10,11-epoxide activity [13]. These changes in the experimental outcomes show the differential mechanisms of CYP3A activities in drug metabolisms. Furthermore, in a 2-week administration study, PJ did not change midazolam PK parameters compared to water intake [46].

Artemether is a drug used to treat malaria, and it is known to be metabolized by CYP450, mainly CYP3A4, CYP1A2, CYP2B6, and CYP2C19 [51,52]. Following a 14-day PJ administration in human volunteers, no significant effect on the pharmacokinetic profile of artemether and its metabolite, dihydroartemisinin, was observed [50].

Although PJ inhibited the rate of 4-OH-flurbiprofen formation from flurbiprofen by human liver microsomes, intake of PJ in humans did not change the PK parameters of flurbiprofen nor flurbiprofen metabolite, 4′-OH-flurbiprofen, in human volunteers compared to placebo and to a potent inhibitor of CYP2C9, fluconazole [16]. In another three-way crossover study, Park et al. (2015) showed that three doses/day for 3 days of PJ did not change simvastatin C_max_ and AUC in comparison to the control period [47].

In an observational study, the co-administration of pomegranate dietary products increased tacrolimus concentration in a patient with heart transplantation [53]. This and other studies led Anlamlert and Sermsappasuk (2020) to study PJ’s effect on cyclosporine pharmacokinetics in healthy volunteers. Cyclosporine, which is a calcineurin inhibitor and plays an important subsiding immunity in transplantation and some autoimmune diseases, has a very narrow therapeutic index. Cyclosporine is metabolized in the intestine and the liver by CYP3A4 and CYP3A5 in humans [54]. Following a single dose of PJ, no change in PK parameters was observed in healthy volunteers, and the differences were within the narrow therapeutic index of cyclosporine [49]. Although Farkas et al. (2007) suggested that the volume in humans may be a reason for the difference between clinical and preclinical studies [14], the latter study used a high volume of pomegranate intake in humans (7.1 mg/kg) and was equivalent to preclinical studies (6.7 mg/kg) [49].

## 6. Conclusions

The above preclinical studies showed that PJ interacts with the intestinal metabolism of CYP3A4 and CYP2C9 enzymes by inhibiting the activity of these enzymes, which results in increasing the bioavailability of CYP3A4 and CYP2C9 mediated drugs (Figure 1). It should be noted, however, that multiple dosing of PJ may have higher interactive potential than a single dose and modulate elimination half-life. Furthermore, PJ has been shown to enhance the absorption of P-glycoprotein absorption-mediated drugs. On the other hand, these effects were not observed on drugs that are not metabolized by CYP3A4 and CYP2C9 enzymes.

In clinical studies, however, the PJ effect on CYP3A and CYP2C9-mediated metabolism was not obvious. Although most of these clinical studies were performed as a single dose of PJ, two clinical studies looked at the impact of 2-week PJ administration. Furthermore, it has been suggested that PJ constituents’ inhibition of the CYP450 enzymes declines within 1–3 days, inferring that ingestion of such medications several hours apart from PJ administration will not prevent the interaction [55]. Other reports have suggested that PJ potential interactions may not be translated into humans [56]. It should be mentioned, however, that the CYP3A4 and CYP2C9 activities are widely variable, and genes variants should be an important factor in humans [57,58]. Furthermore, since CYP3A4 polymorphism is ethnically related [57], more clinical studies are needed to study the effect of PJ on CYP3A4 and CYP2C9-mediated drugs.

Another observation is that the reported drugs in the clinical studies were not the same as those preclinically tested. This opens the need for more clinical studies on PJ’s impact on, for instance, sildenafil, warfarin, carbamazepine, and others. Furthermore, clinical studies should be performed on multi-dosing of PJ to observe the PK and pharmacodynamic impacts better than a single dose, such as those observed in sildenafil and warfarin human observation studies.

## Figures and Tables

**Figure 1 molecules-28-02117-f001:**
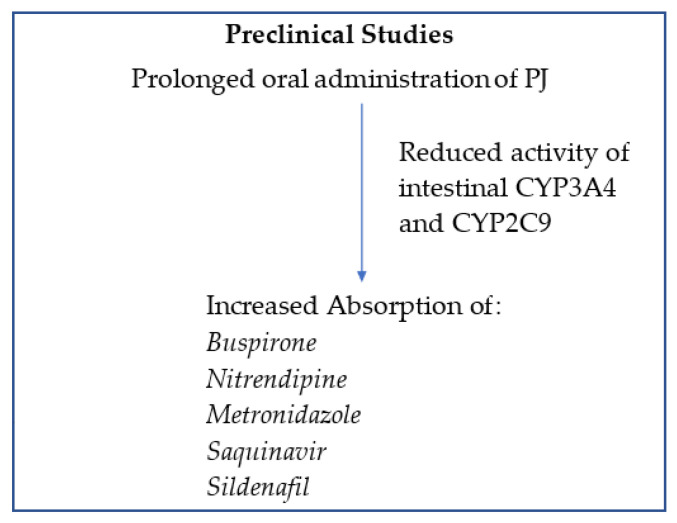
Schematic presentation of prolonged oral administration of PJ on the absorption of drugs metabolized by intestinal CYP3A4 and CYP2C9.

**Table 1 molecules-28-02117-t001:** Preclinical studies investigating the effect of oral administration of pomegranate juice on the PK or PK-PD related to CYP3A4 and CYP2C9-mediated drugs.

Pomegranate Dose	Drug	Animal Species	PK/PD Effects	Drug’s Analytical Technique- Biological Sample *	Reference
Pomegranate juice (2 mL) 1 h before the drug	Carbamazepine (50 mg/kg)	Rats	AUC increased, but no effect on the elimination half-life	HPLC-UV Plasma	[13]
Pomegranate juice (3 mL) administered 1 h before the drug	Tolbutamide (20 mg/kg)	Rats	AUC increased, but no effect on the elimination half-life	HPLC-UV Serum	[15]
Pomegranate juice (10 mL/kg) for 7 days	Buspirone (10 mg/kg)	Rabbits	AUC, C_max_, and elimination half-life increased	HPLC-UV Serum	[17]
Pomegranate juice, single dose Pomegranate juice, multiple doses for 1 week	Nitrendipine (10 mg/kg)	Rabbits	No PK changes AUC and C_max_ increased, but no effect on the elimination half-life	HPLC-UV Plasma	[18]
Pomegranate juice, single dose (3 mL/rat) Pomegranate juice, multiple doses (3 mL/rat/day for 7 days)	Nitrendipine (10 mg/kg)	Rats	AUC and C_max_ increased, but no effect on the elimination half-life AUC and C_max_ increased, but no effect on the elimination half-life	HPLC-UV Plasma	[19]
Pomegranate juice was administered at 5 mL/kg dose 30 min before drug administration or multiple doses twice a day for 2 days	Metronidazole (14 mg/kg)	Rats	No PK changes from a single dose. Multiple doses of Pomegranate increased AUC and C_max_	HPLC-MS Serum	[20]
Pomegranate juice was added to the drinking water 16 h before sildenafil administration, and then each group of rats received 2, 4, 6, and 8 mL of pomegranate juice when sildenafil was administered.	Sildenafil (5 mL/kg)	Rats	AUC increased, and T_max_ delayed	HPLC-UV Plasma	[21]
Pomegranate juice One dose (0.5 mL, 1 mL, and 2 mL/200 g) Dose for 15 days (0.5 mL, 1 mL, and 2 mL/200 g/day)	Saquinavir (100 mg/kg) Saquinavir (100 mg/kg) for 15 days	Rats Rats	AUC, C_max_, and elimination half-life increased AUC, C_max_, and elimination half-life decreased.	HPLC-UV Plasma	[22]
Pomegranate juice was administered as a single dose of 100 mg/kg for 5 days.	Warfarin (0.5 mg/kg)	Rats	Pomegranate did not change PK, but Prothrombin time and INR increased	HPLC-UV Plasma	[23]

* HPLC = High Performance Liquid Chromatography; UV = Ultraviolet; MS = Mass Spectroscopy.

**Table 2 molecules-28-02117-t002:** Preclinical studies investigating the effect of oral administration of pomegranate juice on the pharmacokinetics of drugs not metabolized by CYP3A4 or CYP2C9.

Pomegranate Dose	Drug	Animal Species	PK Effects	Drug Analytical Technique- Biological Sample *	Reference
Pomegranate juice was added to drinking water 12 h before the experiment and a dose of (5 mL) 30 min before the metformin dose.	Metformin (20 mg/kg)	Rats	Pomegranate juice reduced C_max_ (and showed a pattern of AUC reduction)	HPLC-UV Plasma	[37]
Pomegranate juice was administered orally at a dose of 12 mL/kg before the administration of piracetam.	Piracetam (50 mg/kg)	Rats	No change	HPLC-UV Plasma	[38]
Pomegranate juice was added to drinking water 12 h before the experiment and a dose of (5 mL) 30 min before the theophylline dose.	Theophylline (5 mg/kg)	Rats	No change	HPLC-UV Plasma	[39]

* HPLC = High Performance Liquid Chromatography; UV = Ultraviolet.

**Table 3 molecules-28-02117-t003:** Clinical studies investigating the effect of oral administration of pomegranate juice on the pharmacokinetics of CYP3A4 and CYP2C9-mediated drugs.

Pomegranate Dose	Objective Drug	Type of Study	PK Effect	Drug Analytical Technique-Biological Sample(s) *	Reference
Pomegranate juice (200 mL) for 2 weeks	Midazolam and metabolite 1-hydroxymidazolam and 4-hydroxhymidozolam	Open-label, randomized, 2-period, crossover n = 16	No effect on AUC and C_max_	LC-MS/MS Plasma Urine	[46]
Pomegranate juice, one dose of 240 mL or 1 g extract	Flurbiprofen (100 mg)	Open-label, randomized n = 12	No effect on PK	HPLC-FLD Plasma	[16]
Pomegranate juice (237 mL) as a single dose	Midazolam oral (6 mg) Midazolam intravenous (2 mg)	Randomized controlled trial n = 13	No effect on C_max_, AUC, or clearance after oral administration No effect on elimination half-life, volume of distribution, or clearance	LC-MS Plasma	[14]
Pomegranate juice, 3 doses per day (900 mL/day) for 3 days	Simvastatin 40 mg	Open-label, 3-way crossover design n = 12	No significant change in AUC or C_max_.	LC-MS/MS Plasma	[47]
Pomegranate juice (250 mL) for 3 days	Dapoxetine (60 mg) and Midazolam (7.5 mg)	Open-label, 3-way crossover n = 12	Slight but not significant effect on AUC or C_max_	HPLC-UV Plasma	[48]
Pomegranate juice (500 mL)	Cyclosporine (200 mg)	Open-label, randomized, crossover n = 18	No significant change in AUC or C_max_	Chemiluminescent microparticle immunoassay	[49]
Pomegranate juice (250 mL) twice daily doses for 14 days	Artemether (80 mg)	Open-label, randomized, crossover n = 26	No significant change in AUC, C_max,_ T_max_, or elimination half-life for both artemether and dihydroartemisinin (metabolite)	LC-MS/MS Plasma	[50]

* HPLC = High Performance Liquid Chromatography; UV = Ultraviolet; MS = Mass Spectroscopy; FLD = Florescence Detector.

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
