# Peer review of "Impact of Pomegranate Juice on the Pharmacokinetics of CYP3A4- and CYP2C9-Mediated Drugs Metabolism: A Preclinical and Clinical Review"

_molecules, 2023, doi:10.3390/molecules28052117_

Round 1

Reviewer 1 Report

kindly cite following references

1.Khuda, F., Iqbal, Z., Khan, A., Zakiullah, Samiullah, Sahibzada, M. U. K., Alam, M., & Khusro, A. (2021). Effect of fresh pomegranate juice on the pharmacokinetic profile of artemether: An open-label, randomized, 2- period crossover study in healthy human volunteers. Journal of pharmaceutical and biomedical analysis203, 114179. https://doi.org/10.1016/j.jpba.2021.114179

2.El-Sayyad, H. I. H., El-Gallil, H. A., & El-Ghaweet, H. A. (2020). Synergistic effects of pomegranate juice and atorvastatin for improving cerebellar structure and function of breast-feeding rats maternally fed on a high cholesterol diet. Journal of chemical neuroanatomy107, 101798. https://doi.org/10.1016/j.jchemneu.2020.101798

Author Response

Many thanks for your comments.

We included Khuda et al. paper for its high relevance to the review. We included a paragraph stating:

Artemether is a drug used to treat malaria, and it is known to be metabolized by CYP450 mainly CYP3A4, CYP1A2, CYP2B6, CYP2C19 (Ali, Najmi, Tarning, & Lindegardh, 2010; Lefèvre et al., 2002). Following a 14-day PJ administration in human volunteers, no significant effect on the pharmacokinetic profile of artemether and its metabolite, dihydroartemisinin, was observed (Khuda et al., 2021).

Although the El-Sayyad et al. study is very interesting, it is not within the scope of our review.

Reviewer 2 Report

The authors reviewed the preclinical and clinical trials which investigated the effect of pomegranate juice on ADME process of drugs metabolized and non-metabolized by CYP3A4- and CYP2C9 enzymes. I believe that the manuscript has a good level of originality so that a review article with a similar layout and about this theme was never published in another journal, according to a search in the Web of Science database. The main issues of the manuscript are linked to the absence of information about methodological aspects. In line with the latter, also balance and articulation of the methodological approach need to be improved so that the application can be actually demonstrative of the validity of the methodology. The authors did not describe what keywords, time, and databases were used in their review process. Without this information, the search has no reproducibility. Another point to be better explored by the authors is the improvement of criticism of the review article so that some parts of the text are limited to data tabulation. These and some other minor changes are listed below:

(1) Replace carbamazepine 10,11 epoxide, by carbamazepine-10,11-epoxide

(2) In some tables are identified “PK /PD effects” in the column title, however, only PK characteristics are being described.

(3) Check the use of italics in Latin names in all the manuscript (i.e. in vivo) and the text style in some abbreviations such as IC50.

(4) I do not see the necessity of tables in supporting electronic material. The data in electronic supporting material may be merged with the tables found in the main manuscript. Some data in manuscript tables and in supporting information tables are redundant.

(5) may be noticed the absence of a deep criticism level in the discussion of the results. In some parts, the authors are limited to the tabulation and description of the data without their discussion. What the authors think about the quality level of investigations and the future directions of the advances in the field is not reported. Another aspect poorly explored is what are the limitations and challenges of the current investigations. Are there dose and experimental design variations in the current studies? The authors can summarize these data and better explore these points.

Author Response

Many thanks for the constructive comments.

We included a methodology section stating:

Our focus was on the preclinical and clinical studies that investigated the impact of PJ on the PK of drugs that are metabolized by CYP3A and CYP2C9. Therefore, our literature research in electronic databases (PubMed, Scopus, Google, Science Direct) included the following search phrases: “pomegranate and drugs pharmacokinetics”, “pomegranate and CYP3A”, and “pomegranate and CYP2C9”. Any study that was not PK or PK-PD related was excluded. 

The authors highlighted in red, the conclusive statement(s) from each study mentioned in the review. Furthermore, in the conclusion section, we addressed important points on how future clinical studies should be protocoled and what points should be addressed.

As for the minor changes:

  1. Done
  1. The title in the header rows in tables 2 and 3 were changed to “PK effects”.

  1. The file was scanned for any special text styles and corrected (i.e. in vivo, in vitro, IC50, Cmax, Tmax).

  1. Tables (1-3) were merged in the text.

  1. The authors highlighted in red, the conclusive statements from each study mentioned in the review. Furthermore, in the conclusion section, we added the genetic variation in CYP3A4 and CYP2C9 activities in human studies that should be addressed. Furthermore, the important points on how future clinical studies should be protocoled were addressed.

Reviewer 3 Report

This review is very simple and interesting on a good research point. 

I have some comments to improve the work

1- The review did not contain any figures. The author should add one or two figures summarizing the results in a schematic pattern.

2- The author mentioned that pomegranate juice can be used as an anti-cancer, so why he didn't discuss the effect on the metabolism of many anti-cancer drugs metabolized also by cyp3A4 enzymes?

3- Add more references from the literature as there are many other examples of the effect of pomegranate on the metabolism of drugs. 

  • Attwa, M. W.; Kadi, A. A.; Abdelhameed, A. S.; Alhazmi, H. A., Metabolic Stability Assessment of New PARP Inhibitor Talazoparib Using Validated LC-MS/MS Methodology: In silico Metabolic Vulnerability and Toxicity Studies. Drug Des Devel Ther 2020, 14, 783-793.
  • M.W. Attwa, A.A. Kadi, A.S. Abdelhameed, Phase I metabolic profiling and unexpected reactive metabolites in human liver microsome incubations of X-376 using LC-MS/MS: bioactivation pathway elucidation and in silico toxicity studies of its metabolites, RSC Advances, 10 (2020) 5412-5427.
  •  
  •  
  •  

4- You should mention the analytical technique they used to approve the effect on each drug. 

With my best regards:

Author Response

Many thanks for your comments.

  1. A schematic representation has been added to the article to illustrate the effect of prolonged administration of PJ on the absorption of drugs metabolized by intestinal CYP3A4 and CYP2C9.
  2. It would have been a good addition to the manuscript if articles related to the preclinical and/or clinical effect of pomegranate juice on anticancer drugs metabolized by CYP3A4- and CYP2C9 (e.g. vinblastine, docetaxel…etc). Nevertheless, insufficient investigations have been published in this field. Thus, it would serve as a good future research idea!

It is worth mentioning that authors collected many literature recommending the use of PJ during cancer. Though, it was believed they were irrelevant to the context of the review; e.g.

  • Pantuck AJ, Leppert JT, Zomorodian N, Aronson W, Hong J, Barnard RJ, Seeram N, Liker H, Wang H, Elashoff R, Heber D, Aviram M, Ignarro L, Belldegrun A. Phase II study of pomegranate juice for men with rising prostate-specific antigen following surgery or radiation for prostate cancer. Clin Cancer Res. 2006 Jul 1;12(13):4018-26.
  • Paller, C.J., Pantuck, A. and Carducci, M.A., 2017. A review of pomegranate in prostate cancer. Prostate cancer and prostatic diseases, 20(3), pp.265-270.
  • Chaves, F.M., Pavan, I.C.B., da Silva, L.G.S. et al. Pomegranate Juice and Peel Extracts are Able to Inhibit Proliferation, Migration and Colony Formation of Prostate Cancer Cell Lines and Modulate the Akt/mTOR/S6K Signaling Pathway. Plant Foods Hum Nutr 75, 54–62 (2020).
  1. Our focus is on PK-PD correlation of PJ on drugs metabolized by CYP3A and CYP2C9, and not on pomegranate biological effects.
  2. Analytical techniques were added to tables (1-3).

Reviewer 4 Report

This is a nice review summarizing the studies on the pharmacokinetic interactions between pomegranate juice and medications. The topic is of interest and the manuscript relatively well written. A few parts of the manuscript seem to be written a but hastily. Therefore, careful revision is recommended.

Specific comments:

(1) "drug-food interaction became the center of health professionals' interest". A better wording of "center" is maybe "focus".

(2) I don't understand the use of "source" in "Moreover, food-drug interactions are a significant source of medication errors". Maybe consider rewording.

(3) Please consider mentioning which of pomegranate effects were shown in clinical settings, in cell culture, in animal models,...

(4) Different font size in the 2nd paragraph, two dots on the end of the 4th paragraph.

(5) Please correct "cytochrom3" in the 6th paragraph.

(6) Since "PJ" was introduced, please use it (e.g., 1st paragraph of section 2).

(7) I guess "CYP2CA" in the heading of Table 1 should be "CYP2C9".

(8) For the tables, it would me useful to add the information how PJ and the drugs were administered. Always oral administration? Consider also mentioning whether concentrations were measured in plasma, serum or blood.

(9) In section 2.1, "CYP3A" is mentioned, while it is CYP3A4 in the table. The same later in section 4.

(10) What is a "CYP3A metabolism-mediated drug"? Consider rephrasing.

(11) Section 2.6: Consider adding that sildenafil is also used to treat pulmonary hypertension.

(12) Section 2.8. "ratio" not "ration".

(13) Table 2, "Metformin": Generic names are usually lowercase.

(14) Conclusion: It would be interesting to read the author's recommendations. Should patients with specific medications avoid PJ? Or should be a break between medication and PJ administration? If yes, how long? How or when should medications be administered to minimize interactions with food, especially PJ?

Author Response

Many thanks for your comments.

  1. The statement was rephrased as follows:

Hence, drug-food interactions became the focus of health professionals.

  1. The statement has been re-phrased as follows:

Moreover, food-drug interactions might have caused significant medication errors, such as the well-known grapefruit interactions.

  1. Kindly note that the manuscript emphasized on both preclinical and clinical studies. Tables (1,2 and 3) illustrate animal species for preclinical studies as well as clinical studies.
  2. Font size and punctuation have been reviewed throughout the whole document.
  3. “cytochrom3” has been corrected.
  4. “pomegranate juice” has been replaced by “PJ” after defining it in the introduction and abbreviations list.
  5. It has been corrected to CYP2C9.
  6. Pomegranate juice has been orally administered. We added the term ‘Orally” in the heading of the tables.
  7. It has been revised and corrected to CYP3A4.
  8. Rephrasing has been considered.
  9. “Pulmonary hypertension” was added.
  10. “international normalized ratio” was corrected.
  11. “metformin” was corrected.
  12. The authors agree with this suggestion. Nevertheless, it might not be possible to give such advice, taking into consideration that the clinical experimental models were performed as a single dose of PJ except for two studies that looked at the impact of two-week PJ administration. Furthermore, it has been suggested that PJ constituents’ inhibition of the CYP450 enzymes declines within 1-3 days, inferring that ingestion of such medications several hours apart from PJ administration will not prevent the interaction. In addition, the reported drugs in the clinical studies were not the same as those preclinically tested. This opens the need for more clinical studies on PJ impact to be able to give patients such recommendations.

Round 2

Reviewer 2 Report

Thank you for having given me the opportunity to review again the manuscript entitled “Impact of pomegranate juice on the pharmacokinetics of CYP3A4- and CYP2C9-mediated drugs metabolism: A preclinical and clinical review”. I am satisfied with the changes performed by the authors. The manuscript was extensively modified and many aspects were improved in its current version. Thus, in my opinion, the manuscript meets the requirements for publication in Molecules, and I recommend accept it in its current form.

Author Response

Many thanks for your effort.

Reviewer 3 Report

The author did the requested changes, the manuscript could be accepted in it's current format.

With my best regards 

Author Response

Many thanks for your effort.